# The AlkB Homolog SlALKBH10B Negatively Affects Drought and Salt Tolerance in *Solanum lycopersicum*

**DOI:** 10.3390/ijms25010173

**Published:** 2023-12-22

**Authors:** Hui Shen, Ying Zhou, Changguang Liao, Qiaoli Xie, Guoping Chen, Zongli Hu, Ting Wu

**Affiliations:** 1Laboratory of Molecular Biology of Tomato, Bioengineering College, Chongqing University, Chongqing 400030, China; hbshenhui@163.com (H.S.); 15198904590@163.com (Y.Z.); changguangliao@163.com (C.L.); qiaolixie@cqu.edu.cn (Q.X.); chenguoping@cqu.edu.cn (G.C.); 2Key Laboratory of Vegetable Biology of Yunnan Province, College of Landscape and Horticulture, Yunnan Agricultural University, Kunming 650201, China

**Keywords:** AlkB, *SlALKBH10B*, abscisic acid, drought, salt, tomato

## Abstract

ALKBH proteins, the homologs of *Escherichia coli* AlkB dioxygenase, constitute a single-protein repair system that safeguards cellular DNA and RNA against the harmful effects of alkylating agents. ALKBH10B, the first discovered *N*^6^-methyladenosine (m^6^A) demethylase in Arabidopsis (*Arabidopsis thaliana*), has been shown to regulate plant growth, development, and stress responses. However, until now, the functional role of the plant ALKBH10B has solely been reported in arabidopsis, cotton, and poplar, leaving its functional implications in other plant species shrouded in mystery. In this study, we identified the AlkB homolog SlALKBH10B in tomato (*Solanum lycopersicum*) through phylogenetic and gene expression analyses. *SlALKBH10B* exhibited a wide range of expression patterns and was induced by exogenous abscisic acid (ABA) and abiotic stresses. By employing CRISPR/Cas9 gene editing techniques to knock out *SlALKBH10B*, we observed an increased sensitivity of mutants to ABA treatment and upregulation of gene expression related to ABA synthesis and response. Furthermore, the *Slalkbh10b* mutants displayed an enhanced tolerance to drought and salt stress, characterized by higher water retention, accumulation of photosynthetic products, proline accumulation, and lower levels of reactive oxygen species and cellular damage. Collectively, these findings provide insights into the negative impact of *SlALKBH10B* on drought and salt tolerance in tomato plant, expanding our understanding of the biological functionality of *SlALKBH10B*.

## 1. Introduction

Approximately four decades ago, researchers discovered that the AlkB protein possesses the extraordinary ability to function as a solitary repair system, safeguarding *E. coli* from the detrimental effects of alkylating agents [1]. In the presence of nonheme Fe(II) and requisite cofactors, AlkB exhibits oxidative demethylation of DNA/RNA bases, restoring the original structure of nucleic acids [2,3,4]. While prokaryotes typically possess one to three AlkB proteins, eukaryotes, such as homo sapiens and *Arabidopsis thaliana*, boast a more extensive repertoire, with nine and thirteen homologs of AlkB (ALKBH), respectively [5,6]. Moreover, eukaryotic ALKBH proteins participate in many crucial cellular regulatory processes, encompassing the repair of alkylation-induced damage in DNA, RNA, and nucleoprotein complexes [5]. Compared to their prokaryotic ancestors, this expanded array of functionalities underscores the versatility of eukaryotic ALKBH proteins. It is worth highlighting that the catalytic prowess of specific ALKBH proteins, namely FTO and ALKBH5, in the removal of *N*^6^-methyl adenosine (m^6^A) from mRNA has engendered considerable intrigue in recent years [7,8], rendering them a captivating subject of scientific inquiry.

As early as the 1970s, researchers detected the presence of m^6^A modification in the mRNA of mammalian cells [9]; however, its biological significance remained enigmatic. The elucidation of m^6^A’s role in mRNA modification did not occur until 2008, when scholars observed that the absence of MTA, a constituent of the m^6^A methyltransferase complex, resulted in embryonic lethality in *A. thaliana* [10]. In 2011, it was discovered that the obesity-associated protein (FTO), a member of the ALKBH family, served as an m^6^A demethylase for mRNAs [7]. This finding unveiled the reversible nature of m^6^A modification and its pivotal involvement in regulating cellular gene expression. Subsequently, it was confirmed that another member of the ALKBH family, human ALKBH5, possessed m^6^A demethylase activity [8]. While plants lack a homologue of FTO, they do have multiple homologues of ALKBH5. For instance, within the *Arabidopsis* genome, thirteen ALKBH family proteins are encoded, with five (ALKBH9A/9B/9C/10A/10B) residing within the ALKBH9 and ALKBH10 subfamilies, exhibiting homology to ALKBH5 [6,11]. Notably, ALKBH9B and ALKBH10B have been experimentally shown to possess m^6^A demethylase activity, exerting influence over m^6^A modification in vivo [12,13].

The plant ALKBH10, a homolog of α-ketoglutarate-dependent dioxygenase, was initially discovered as the first m^6^A demethylase in *A. thaliana*. Specifically, AtALKBH10B has been demonstrated to remove m^6^A modifications on RNA, both in vivo and in vitro, and the *alkbh10b* mutants exhibited a distinct delay in their vegetative growth and reproductive development [13]. These findings strongly support the role of ALKBH10B as an m^6^A demethylase. In addition to its involvement in plant growth and development, recent studies have revealed the role of *ALKBH10B* in regulating abiotic stress responses in plants. In *Arabidopsis*, the *alkbh10b* mutant exhibited a salt-tolerant phenotype [14]. Several genes, acting as negative regulators of salt stress tolerance, were identified as m^6^A-modified genes, and their expression levels were decreased in the *alkbh10b* mutant [14]. Furthermore, in response to abscisic acid (ABA), the *alkbh10b* mutant inhibited seedling and root growth by up-regulating ABA signaling-related genes *ABI3* and *ABI4* [14]. Additionally, *GhALKBH10* has been identified as a negative regulator of salt and drought tolerance in cotton [15,16]. However, it is essential to note that the biological function of *ALKBH10B* in other plant species is still unknown.

Tomato (*Solanum lycopersicum*), a prominent cash crop and a model plant for investigating plant growth and development manifests a regulatory interplay between the m^6^A demethylase SlALKBH2 and the DNA demethylase gene *SlDML2*. Specifically, SlALKBH2 reduces the m^6^A modification level of the *SlDML2* transcript through m^6^A demethylation, ultimately leading to stabilize *SlDML2* mRNA [17]. Mutations in *SIALKBH2* decrease the abundance of *SlDML2* mRNA, consequently leading to delayed tomato fruit ripening [17]. In addition, SIALKBH2 was classified as an ALKBH9 subfamily protein in a phylogenetic analysis of the tomato ALKBH family [18]. However, the functions of tomato ALKBH10 subfamily proteins remain elusive.

In this study, a member of the ALKBH10 subfamily protein was identified in tomato, denoted as SlALKBH10B, which exhibited homology with *Arabidopsis* ALKBH10B and human ALKBH5. Drawing upon the known biological function of ALKBH10B in plants, we hypothesized that SlALKBH10B plays a pivotal role in the tomato plant’s response to abiotic stress. To put this hypothesis to the test, we employed the CRISPR/Cas9 technology to generate *Slalkbh10b* mutant lines. Remarkably, our findings revealed that the mutant lines exhibited heightened sensitivity to exogenous ABA treatment. Furthermore, we have observed a remarkable enhancement in the drought tolerance and salt stress resilience of the *Slalkbh10b* mutant at the seedling stage. The phenotypic alterations were confirmed through morphological, physiological, and molecular analyses of *Slalkbh10b* mutant lines. These research findings provide invaluable insights into the functional role of *SlALKBH10* in alleviating the impacts of drought and salt stress, thereby illuminating its potential significance in enhancing plant resilience.

## 2. Results

### 2.1. SlALKBH10B Is a Homologue of HsALKBH5

To investigate the evolutionary relationship of ALKBH family proteins in tomato (*Solanum lycopersicum*), we constructed a phylogenetic tree using the protein sequences of ALKBH from tomato, Arabidopsis, and humans. Based on the homology shared with human ALKBH proteins, the ALKBH family proteins in tomato can be classified into five distinct subfamilies, namely ALKBH1, ALKBH2, ALKBH5, ALKBH6, and ALKBH8 (Figure 1). In addition, like Arabidopsis, tomato lacks any FTO homologues, yet it possesses multiple homologues of HsALKBH5. In tomato, five HsALKBH5 homologs (SlALKBH9A/B/C and SlALKBH10A/B) can be subdivided into two subbranches, namely ALKBH9 and ALKBH10 (Figure 1). Among these, SlALKBH9A, also known as SlALKBH2, has been experimentally verified as an m^6^A demethylase that plays a pivotal role in regulating of fruit ripening [17]. AtALKBH9B and AtALKBH10B are the most extensively investigated m^6^A demethylases in the plant kingdom, orchestrating various aspects of plant growth, development, and stress responses. In tomato, our phylogenetic analysis found that SlALKBH10B is a homologue of AtALKBH10B and HsALKBH5 (Figure 1), thereby harboring the potential for m^6^A demethylase activity. Nonetheless, the precise biological function of SlALKBH10B remains shrouded in mystery.

### 2.2. SlALKBH10B Exhibits Broad Tissue Expression and Responds to Hormones and Abiotic Stress Treatments

To explore the gene expression levels of five ALKBH5 homologues in tomato plant, we executed a comprehensive gene expression heatmap analysis employing transcriptome data (Tomato genome construction, 2012). The results revealed that *SlALKBH9B*, also known as *SlALKBH2*, exhibits a distinctive and robust expression, specifically during the fruit ripening process (Figure 2A), aligning with its established role in regulating this physiological phenomenon [17]. Conversely, *SlALKBH9B*, *SlALKBH9C*, and *SlALKBH10A* exhibit low expression levels across various tissues (Figure 2A). In contrast, *SlALKBH10B* manifests ubiquitous expression across all tissues (Figure 2A), indicating its pervasive regulatory influence in the growth and development of tomato plant. Further validation through quantitative reverse transcription-polymerase chain reaction (qRT-PCR) analysis corroborated the expression of *SlALKBH10B* across all tomato tissues, with relatively higher expression levels detected in young leaves, flowers, and fruits (Figure 2B). Additionally, the results of qRT-PCR analysis unveiled that the expression of *SlALKBH10B* can be induced by the exogenous abscisic acid (ABA) and gibberellin (GA_3_) (Figure 2C,D). Conversely, the expression of *SlALKBH10B* remains unaffected mainly by auxin (IAA) and salicylic acid (SA) treatment (Figure 2E,F). Under abiotic stress’s duress, both drought and salt stress emerge as potent instigators of heightened *SlALKBH10B* expression (Figure 2G,H). These findings underscore that *SlALKBH10B* likely assumes a multifaceted role in tomato growth, development, and abiotic stress (drought and salt) responses.

### 2.3. SlALKBH10B Localizes to the Cytoplasm and Nucleus

Previous investigations have established that ALKBH5 homologues in mammals and plants exhibit divergent subcellular localizations. HsALKBH5 is localized in the nucleus [19], while AtALKBH9B and SlALKBH9B are cytoplasmic proteins [12,17]. Conversely, AtALKBH10B assumes a nuclear-cytoplasmic distribution pattern [6]. Transient expression experiments were conducted in tobacco leaves to ascertain the subcellular localization of SlALKBH10B, employing HY5-RFP as a nuclear localization signal. The green fluorescent signals were observed in the control group in the nucleus and cytoplasm (Figure 3A). Simultaneously, the green fluorescent signals emitted by the SlALKBH10B-YFP fusion protein also exhibited a nuclear-cytoplasmic distribution pattern (Figure 3B). These findings corroborate that SlALKBH10B localizes to the nucleus and cytoplasm, aligning with the subcellular distribution observed in its Arabidopsis homologue, AtALKBH10B.

### 2.4. Knockout of SlALKBH10B Exhibited Heightened Sensitivity to ABA

By employing genetic editing techniques, genetic transformation, and genomic sequencing, we successfully generated three distinct genotypes of homozygous *Slalkbh10b* mutant lines, namely *CR-15*, *CR-38*, and *CR-46* (Figure 4A and Appendix A). These base insertions and deletions in the first exon of *SlALKBH10B* resulted in a frameshift mutation in the coding sequence, ultimately leading to translation termination. Given that the exogenous ABA can induce the expression of *SlALKBH10B* (Figure 2C), it is postulated that *SlALKBH10B* may participate in ABA synthesis or signaling pathways. Upon exogenous ABA treatment, the growth of *Slalkbh10b* mutant lines was notably inhibited, with observable effects such as thinner and purplish hypocotyls (Figure 4B). Statistical analysis further revealed that the root length and hypocotyl length of *Slalkbh10b* mutant lines were significantly shorter than the wild-type (WT) under 2 µM and 4 µM ABA treatments (Figure 4C,D). These results suggested that the *Slalkbh10b* seedlings exhibit heightened sensitivity to exogenous ABA treatment, indicating a potential increase in endogenous ABA content. Moreover, subsequent qRT-PCR analysis demonstrated that the expression levels of *NCED1* (a rate-limiting enzyme gene in ABA synthesis), as well as *ABI3* and *ABI5* (genes responsive to ABA), were significantly elevated in the *Slalkbh10b* lines under 2 µM ABA treatment (Figure 4E–G). Consequently, the knockout of *SlALKBH10B* resulted in enhanced expression levels of ABA synthesis and response genes, ultimately amplifying the sensitivity of *Slalkbh10b* seedlings to ABA treatment.

### 2.5. Knockout of SlALKBH10B Improved Drought Tolerance

Given the notable upregulation of *SlALKBH10B* expression in response to drought stress (Figure 2G), coupled with the heightened susceptibility of *Slalkbh10b* mutants to ABA (Figure 4B), we embarked upon subjecting both the WT and *Slalkbh10b* mutant plants to drought stress treatment. Before the commencement of the drought treatment, no significant difference in the growth status was observed between the WT and *Slalkbh10b* plants (Figure 5A). However, on the fifteenth day of the drought treatment, the WT plants showed difficulty supporting, falling, and drooping, with most leaves wilting (Figure 5A). Conversely, the *Slalkbh10b* plants maintained an upright posture, with only the basal leaves turning yellow and wilting and the remaining leaves in better condition (Figure 5A). Water-related index tests showed that the water loss rate of *Slalkbh10b* leaves was lower than that of WT (Figure 5B). Furthermore, the relative water content after 15 days of drought treatment was significantly higher in *Slalkbh10b* leaves than in WT (Figure 5C). The photosynthesis-related indices revealed that the levels of chlorophyll (Figure 5D), soluble sugar (Figure 5E), and starch (Figure 5F) in *Slalkbh10b* leaves were significantly higher than those in WT after 15 days of drought treatment. Moreover, after 15 days of drought treatment, the expression levels of the dihydrodipicolinate reductase-like gene, *CRR1* (Figure 5G) and fructokinase gene, *FRK2* (Figure 5I), two positive regulators of chloroplast development and glycolysis [20,21,22], were significantly higher in *Slalkbh10b* leaves compared to WT. Conversely, the expression levels of senescence-inducible chloroplast stay-green protein 1, *SGR1* (Figure 5H), a negative regulator of chlorophyll degradation [23,24], were significantly lower in *Slalkbh10b* leaves compared to WT. These findings indicated that under drought stress, *Slalkbh10b* plants exhibit enhanced water utilization and photosynthetic capacity compared to WT plants, thereby displaying an increased tolerance to drought stress.

### 2.6. Knockout of SlALKBH10B Reduces Cell Damage after Drought Stress

After plants suffer from adverse stress conditions, their cells may exhibit varying degrees of damage, with indicators such as proline and malondialdehyde (MDA) content, relative electric conductivity (REC), and hydrogen peroxide (H_2_O_2)_ levels. Before the drought treatment, there were no significant differences in these physiological indices between the leaves of WT and *Slalkbh10b* plants. However, following a 15-day period of drought treatment, the proline content (Figure 6A) in the leaves of *Slalkbh10b* was significantly higher than that in WT. In contrast, the levels of MDA (Figure 6B), H_2_O_2_ (Figure 6C), and REC (Figure 6D) were significantly lower in *Slalkbh10b* compared to the WT. To gain further information concerning the increased drought tolerance phenotype observed in *Slalkbh10b* plants, transcripts of previously reported stress-related genes were used to compare WT and *Slalkbh10b* leaves under normal and drought-stressed conditions. Two cytosolic ascorbate peroxidase (APX) genes, *APX1* and *APX2* [25]; two catalase (CAT) genes, *CAT1* and *CAT2* [26]; one superoxide dismutase (Cu/ZnSOD1) gene [27], and the proline synthesis gene (*P5CS*) [28] were selected (Figure 6E–J). Quantitative RT-PCR analysis showed that the transcripts of various antioxidant enzyme-related genes (*APX1*, *APX2*, *CAT1*, *CAT2*, and *Cu/ZnSOD1*), as well as the proline synthesis gene (*P5CS*) in the leaves of *Slalkbh10b*, were significantly elevated after the 15-day drought treatment, surpassing the levels observed in the WT (Figure 6E–J). Moreover, the results of trypan blue staining indicated that the blue area on the leaves of *Slalkbh10b* was minor, and the number of viable cells was higher than the WT (Figure 6K). These findings suggested that, in the face of drought stress, *Slalkbh10b* plants exhibit a reduced level of cell damage compared to the WT.

### 2.7. Knockout of SlALKBH10B Enhances Salt Tolerance

Considering the substantial upregulation of *SlALKBH10B* expression in response to salt treatment (Figure 2H), we subjected both the WT and *Slalkbh10b* plants to salt stress experiments. Following a 20-day duration of salt stress treatment, the WT plants exhibited pronounced leaf abscission, with only a few uppermost leaves remaining, while the majority wilted (Figure 7A). In contrast, the *Slalkbh10b* plants maintained an upright morphology, manifesting only slight yellowing and wilting of the basal leaves, while the remainder remained robust (Figure 7A). Furthermore, after the salt stress treatment, the relative water content of the *Slalkbh10b* leaves was notably higher than that of the WT (Figure 7B). The indices associated with photosynthesis unveiled that the *Slalkbh10b* mutants displayed significantly elevated levels of chlorophyll (Figure 7C), soluble sugar (Figure 7D), and starch (Figure 7E) content in comparison to the WT following the 20-day salt stress treatment. Moreover, a subsequent qRT-PCR analysis revealed an upregulation in the expression levels of genes associated with chloroplast development, namely *Golden2-like2* [29] and *CRR1* [20] (Figure 7F,G), as well as the sugar metabolism-related gene *FRK2* [21,22] (Figure 7H), within the leaves of *Slalkbh10b* relative to the WT. These findings proposed that under salt stress conditions, *Slalkbh10b* plants possess a heightened ability to harness water and engage in photosynthesis, consequently exhibiting heightened tolerance to salt stress.

### 2.8. Knockout of SlALKBH10B Reduces Cell Damage after Salt Stress

The imposition of salt stress treatment also elicits cellular impairment in plants, resulting in inevitable increments in proline, MDA, H_2_O_2_ content, and REC (Figure 8A–D). Nevertheless, after a 20-day duration of salt stress treatment, the leaves of *Slalkbh10b* plants exhibited a significantly augmented proline content in contrast to the WT (Figure 8A). In contrast, the levels of MDA (Figure 8B), H_2_O_2_ (Figure 8C), and REC (Figure 8D) were notably diminished in the *Slalkbh10b* plants relative to the WT. These physiological markers align with the observed ameliorated salt stress tolerance in the *Slalkbh10b* plants. Similarly, two cytosolic ascorbate peroxidase (APX) genes, *APX1* and *APX2* [25]; one catalase (CAT) genes, *CAT1* [26]; one superoxide dismutase (Cu/ZnSOD1) gene, *Cu/ZnSOD1* [27]; the proline synthesis gene, *P5CS* [28], and the salt-related dehydrin gene, *TAS14* [30], were selected for comparative analysis between WT and *Slalkbh10b* leaves under normal and salt-stressed conditions (Figure 8E–J). Analysis of gene expression unveiled that the expression levels of various genes associated with antioxidant enzymes (*APX1*, *APX2*, *CAT2*, and *Cu/ZnSOD1*), proline synthesis (*P5CS*), and stress response (*TAS14*) were significantly upregulated in the leaves of *Slalkbh10b* plants compared to the WT (Figure 8E–J). Furthermore, trypan blue staining revealed a reduced blue area on the leaves of *Slalkbh10b* plants, and the number of viable cells was higher than the WT (Figure 8K). These findings suggest that the *Slalkbh10b* plants experience diminished cellular damage under salt stress conditions when juxtaposed with the WT.

## 3. Discussion

Research on ALKBH proteins in botany remains relatively scarce. Existing studies primarily focused on homologs of ALKBH5 in *A*. *thaliana*. Phylogenetic analysis has revealed the absence of FTO homologs in plants, while multiple homologs of ALKBH5 can be found [11,31]. Consistent with these findings, our phylogenetic analysis indicated that tomato (*Solanum lycopersicum*) lack FTO homologs, but possess five homologs of ALKBH5 (Figure 1). Furthermore, tomato and *A*. *thaliana* exhibit further subdivisions within the ALKBH5 homologs resulted in two branches: ALKBH9 and ALKBH10 (Figure 1). Notably, both AtALKBH9B and AtALKBH10B in *A. thaliana* have been confirmed as m^6^A demethylases, yet they possess distinct biological functions. The RNA m^6^A demethylase AtALKBH9B plays a role in the process of *A. thaliana* infection by the alfalfa mosaic virus by removing m^6^A modifications from its RNA [12]. However, single mutants in *alkbh9b* did not exhibit significant changes in endogenous mRNA m^6^A levels [13]. In contrast, disruption of *AtALKBH10B* led to global m^6^A hypermethylation, resulting in delayed flowering and suppressed vegetative growth [13]. These observed differences in their biological characteristics may contribute to the functional disparities observed. For instance, ALKBH9B localizes to cytoplasmic bodies [6,12], whereas ALKBH10B can be found in the cytoplasm and nucleus [6]. In this study, among the five homologs of ALKBH5 encoded by tomato genes, only *SlALKBH9A* and *SlALKBH10B* exhibited higher expression levels (Figure 2A). Specifically, SlALKBH9A served as a ripening-specific regulator, exerting its influence on the stability of the transcript of the DNA methyltransferase *SlDML2* through its m^6^A demethylase activity [17]. In comparison, *SlALKBH10B* displayed a broader tissue expression pattern (Figure 2A, B), suggesting its involvement in regulating growth and development at various stages of tomato development. Additionally, subcellular localization analysis revealed that SlALKBH9A is located in the endoplasmic reticulum (ER) [17], while our research indicated that SlALKBH10B is found in both the cytoplasm and nucleus (Figure 3). The subcellular localization of ALKBH10B in plants appears to be conserved, as evidenced by the simultaneous nuclear and cytoplasmic localization of GhALKBH10B in cotton [15].

In recent years, there has been more in-depth research on the ALKBH10B protein in a few plant species, revealing its impact on plant growth, flowering transition, and its pivotal role in responding to abiotic stress. Notably, *ALKBH10B* exhibits diverse responses to abiotic stresses and ABA treatment in terms of gene expression. For instance, in *Arabidopsis*, the expression of *ALKBH10B* was observed to be upregulated in response to ABA, osmotic stress, salt stress, and drought stress [14,32]. Conversely, in cotton, *GhALKBH10B* was found to be downregulated under different abiotic stresses [15]. Similarly, in *Populus*, salt treatment induced the expression of *PagALKBH10B* [33]. In this study, we observed significant upregulation of *SlALKBH10B* expression in response to ABA, drought, and salt treatments (Figure 2C,G,H). ABA, a plant hormone involved in various abiotic stress responses, facilitates plant adaptation to adverse environmental conditions [34]. Research has demonstrated that *ALKBH10B* influences plant sensitivity to ABA and modulates gene expression levels in the ABA signaling pathway. In *Arabidopsis*, *alkbh10b* mutants exhibited heightened sensitivity to ABA, osmotic, and salt stress during seed germination [14]. These mutants also exhibited reduced seedling and root growth in response to ABA treatment [32]. In cotton, the m^6^A demethylase GhALKBH10B decreased m^6^A levels, thereby facilitating the mRNA decay of ABA signal-related genes [16]. These findings suggest that ALKBH10 negatively regulates ABA signaling in *Arabidopsis* and cotton. In this study, the *Slalkbh10b* mutants exhibited increased sensitivity to a 7-day periodic exogenous ABA treatment compared to the WT, and the inhibition of hypocotyl elongation and root growth in the *Slalkbh10b* mutants was more severe after ABA treatment (Figure 4B–D). Gene expression analysis revealed significant upregulation of ABA synthesis (*NCED1)* and response genes (*ABI3* and *ABI5*) in the *Slalkbh10b* mutants following ABA treatment (Figure 4E–G). Yet, the precise nature of the early response signals remains elusive, owing to the adaptability challenges posed by the 7-day duration. Nonetheless, in line with the functional role of *ALKBH10B* in *Arabidopsis* and cotton, our results suggest that *SlALKBH10B* acts as a negative regulator of the ABA-signaling pathway in tomato.

The involvement of ALKBH10B in salt and drought stress regulation has been documented. However, the regulatory mechanisms of ALKBH10B in salt and drought tolerance vary among plant species. For instance, in *Arabidopsis*, loss-of-function mutants of *ALKBH10B* exhibited enhanced salt tolerance [14], but they were sensitive to drought stress [32], suggesting a negative regulation of salt tolerance and a positive regulation of drought tolerance by AtALKBH10B. In cotton, silencing *GhALKBH10B* led to improved salinity tolerance [15], and mutation of *GhALKBH10B* enhanced drought resistance during the seedling stage [16], indicating a negative regulation of both salt and drought tolerance by GhALKBH10B. Moreover, heterologous expression of *PagALKBH10B* in *Arabidopsis* resulted in increased salt tolerance but decreased drought tolerance in transgenic lines [33], implying a positive regulation of salt tolerance and a negative regulation of drought tolerance by PagALKBH10B. These findings highlight the diverse and even opposing biological functions of ALKBH10B in regulating salt and drought tolerance across different plant species. Further elucidation of the specific regulatory pathways involving ALKBH10B in salt and drought stress is required. In this study, the knockout of *SlALKBH10* improved drought and salt tolerance in tomato (Figure 6A and Figure 8A), suggesting that *SlALKBH10* in tomato shares a similar function with *GhALKBH10B* in cotton and acts as a negative regulator of both drought and salt tolerance.

Following drought and salt stress, plants undergo various physiological changes, including alterations in water content and photosynthetic efficiency, and the accumulation of reactive oxygen species and cellular damage. The water loss rate and detached leaves’ relative water content indicate plant drought tolerance [35]. Plants exhibiting lower water loss rates and higher relative water content demonstrate enhanced drought tolerance. Chlorophyll, soluble sugars, and starch levels can indicate photosynthetic efficiency, with *CRR1* playing a positive regulatory role in chloroplast development [20]. *SGR1* is involved in chlorophyll degradation [23,24], while *FRK2* exerts a positive regulatory effect on sugar metabolism [21,22]. Cellular damage can be assessed through indicators, such as relative electric conductivity (REC), malondialdehyde (MDA), and hydrogen peroxide (H_2_O_2_). REC indicates membrane permeability, with higher values suggesting increased membrane permeability and more severe cellular damage. MDA is the end product of membrane lipid peroxidation [36], and its excessive production exacerbates membrane damage. H_2_O_2_ is associated with reactive oxygen metabolism, with higher levels indicating more significant plant oxidative damage. Additionally, plants produce specific substances to protect themselves during stress, such as the synthesis of proline to maintain cellular structural stability and eliminate internal reactive oxygen species. Genes such as *CAT1*, *CAT2*, *APX1*, *APX2*, and *SOD* play a role in eliminating oxidative damage through the antioxidant enzyme pathway [25,26,27], while *P5CS* is involved in proline synthesis [28].

In this study, the knockout of *SlALKBH10B* in tomato plants confer augmented resilience against drought and salinity during the seedling stage. In a natural setting, the leaves of *Slalkbh10b* seedlings exhibited diminished water loss rates compared to WT (Figure 5B). Following exposure to drought and salinity treatments, the *Slalkbh10b* seedlings displayed a higher relative water content in their leaves than the WT (Figure 5C and Figure 7B). Moreover, an elevation in proline levels was observed in the *Slalkbh10b* seedlings (Figure 6A and Figure 8A), accompanied by the upregulation of genes associated with proline synthesis (Figure 6J and Figure 8I). Additionally, the *Slalkbh10b* seedlings displayed heightened levels of chlorophyll content, soluble sugars, and starch (Figure 5D–F and Figure 7C–E), coupled with the upregulated of genes related to chlorophyll synthesis, the downregulation of degradation-related genes, and the upregulation of genes involved in sugar metabolism (Figure 5G–I and Figure 7F–H). Regarding cellular damage, the *Slalkbh10b* seedlings demonstrated reduced levels of MDA (Figure 6B and Figure 8B), H_2_O_2_ (Figure 6C and Figure 8C), and REC (Figure 6D and Figure 8D), as well as lighter trypan blue staining (Figure 6J and Figure 8J). Furthermore, the expression of genes associated with antioxidant enzymes was upregulated in the *Slalkbh10b* seedlings (Figure 6E–I and Figure 8E–H). Collectively, these findings suggest that, in comparison to the WT, the *Slalkbh10b* seedlings possess an enhanced ability to effectively utilize and retain water, synthesize protective agents, accumulate organic compounds, and express antioxidant enzymes, thereby bolstering the plant’s resistance to adverse conditions encountered during drought and salt stress. Certainly, a comprehensive understanding of the precise regulatory mechanisms underlying the role of *SlALKBH10B* in drought and salinity stress necessitates further investigation. This study sheds light on the biological significance of the AlkB homolog, SlALKBH10B, in conferring tolerance to abiotic stress in tomato, thereby contributing to our knowledge of ALKBH10B in horticultural crops.

## 4. Materials and Methods

### 4.1. Plant Materials and Growth Conditions

*Solanum lycopersicum* cv. Ailsa Craig plants, including wild-type (WT) and transgenic mutant lines (*SlALKBH10B*-CRISPR/Cas9), were cultivated in a controlled greenhouse under 16 h of daylight at 27 °C and 8 h of night time at 19 °C, with relative humidity maintained at 80%. Samples of roots, stems, flowers, leaves, petioles, inflorescence, sepals and fruits were collected from WT plants for organ-specific expression profiles. Labels were used to mark the flowers during anthesis to determine the timing of fruit development and ripening as days post-anthesis (DPA). At 25 and 35 DPA, fruits were classified as immature green (IMG) and mature green (MG), respectively. The onset of fruit color change serves as an indicator for the breaker (B) stage. The ripening periods were classified as four days after breaker (B + 4) and seven days after breaker (B + 7). All biological materials were immediately frozen in liquid nitrogen and stored at –80 °C.

### 4.2. Evolutionary Analysis of ALKBH Family Proteins in Tomato

To analyze the phylogenetic evolution of ALKBH family proteins in tomato, we employed MAFFT v7 [37] to align selected full-length amino acid sequences, utilizing default parameters. Subsequently, the resulting alignment was utilized to construct a neighbour-joining (NJ) tree using MEGA11 [38], employing Poisson correction, partial deletion, and 1000 bootstrap replicates. To enhance the visual representation and facilitate distinction within the phylogenetic tree, we used the interactive tree of life (iTOL) v6.3 (https://itol.embl.de/ (accessed on 19 August 2021)), which enabled us to apply coloration to various branches. To ensure consistency and clarity, we referred to previous research for the sequences of ALKBH family proteins in tomato, Arabidopsis, and human. Subsequently, we assigned new names based on our phylogenetic analysis. The Appendix A, Appendix A provides comprehensive information regarding the reference sequences and the corresponding renaming.

### 4.3. Expression Analysis of SlALKBH10B

To conduct a comparative analysis of the expression levels of tomato ALKBH9 and ALKBH10 subfamily genes in different tissues, we obtained RNA-Seq data from a previously published study on tomato genome construction (2012). The acquired RNA-Seq data was normalized using log2 (reads per kilobase of per million mapped reads (RPKM)) values and subsequently visualized to generate a heatmap. This visualization was accomplished using the OmicStudio tools (https://www.omicstudio.cn/tool/ (accessed on 28 August 2021)). Furthermore, the relative expression of *SlALKBH10B* in various tomato tissues was assessed through qRT-PCR analysis. Treatment assays were performed according to our previous study [39], and WT tomato plants approximately 1-month-old were utilized. In the case of phytohormone treatment, the tomato plants were sprayed with 100 μM ABA, 50 μM SA, 50 μM GA3, and 50 μM IAA until droplets formed on the leaves. Sterile water was employed as a control. Leaf samples were collected at specific time points 0, 1, 2, 4, 8, 12, and 24 h after treatment. Regarding the dehydration treatment, the tomato seedlings were carefully uprooted, and the soil adhering to the roots was gently removed. Subsequently, all the tomato seedlings were placed on dry filter paper. As for the salt treatment, the roots of the tomato seedlings were exposed to a 200 mM NaCl solution. Leaf samples were collected at designated time points, specifically 0, 2, 4, 8, 12, and 24 h after treatment. Three biological replicates were conducted. All collected samples were promptly wrapped in foil, flash-frozen using liquid nitrogen, and stored in a refrigerator at −80 °C for subsequent analysis.

### 4.4. Subcellular Localization

To generate the SlALKBH10B-YFP fusion protein, the coding sequence (CDS) fragment of *SlALKBH10B*, excluding the stop codon, was subjected to *BamH* I and *Sac* I digestion. The digested fragment was then inserted into the pHB-YFP vector. Agrobacterium tumefaciens strains GV3101, containing pHB-SlALKBH10B-YFP and the control vector pHB-YFP, were mixed separately with the GV3101 strain containing HY5-RFP (each strain, OD 600 = 1.0, 1:1 by volume). The mixture obtained was subsequently infiltrated into 4-week-old tobacco (*N. benthamiana*) leaves. Direct observations of the tobacco leaves were carried out two days following infiltration, and images were captured using a laser confocal microscope (Leica TCS SP8). The excitation wavelengths were 488 nm for GFP and 563 nm for RFP, and the emission wavelengths were 507 nm for GFP and 582 nm for RFP. The HY5-RFP was used as described previously [40].

### 4.5. Generation of SlALKBH10B-CRISPR/Cas9 Transgenic Plants

To target *SlALKBH10B* in tomato, the CRISPR-P tool (http://cbi.hzau.edu.cn/crispr/ (accessed on 30 August 2021)) was employed to design single-guide RNAs (sgRNAs). The pKSE-401 binary vector [41] was utilized for the implementation of a single-target knock-out strategy for *SlALKBH10B*. As previously described, the recombinant knock-out vector was introduced into WT plants through a stable transformation using LBA4404 [42]. Transgenic lines were selected using MS medium supplemented with kanamycin (50 mg/mL), and their presence was confirmed through PCR using NPTII-F/R primers. Genomic DNA was extracted from young leaves of both the WT plants and the transgenic lines using a Genomic DNA Extraction Kit (Invitrogen, Shanghai, China). The target genomic sequences were amplified through PCR using primers that flank the target sites. The resulting PCR products were subsequently sequenced to identify any sequence mutations. Furthermore, the potential off-target effects were predicted using the CRISPR-P tool, and these predictions were further validated through cloning and sequencing. Primers used to analyze target and off-target site mutations can be found in Appendix A and Appendix A, respectively. The detection of mutations in putative off-target sites is presented in Appendix A.

### 4.6. ABA Sensitivity Analysis

The WT and *Slalkbh10b* mutant seeds were collected under identical environmental conditions. To ensure sterility, the seeds were subjected to a series of surface sterilization steps, including a 1-min treatment with 70% ethanol and a 5-min treatment with 20% sodium hypochlorite. The sterilized seeds were then rinsed with distilled water and then sown on 1/2 MS medium (added 3% sucrose by weight) supplemented with different concentrations of ABA, namely 0 μM ABA, 2 μM ABA, or 4 μM ABA. After seven days of cultivation in a light incubator (Ningbo Jiangnan Instrument Factoy (Ningbo, China), GXZ-1000B, 15,000 lux), the lengths of the roots and hypocotyls of the seedlings were measured. Samples of seedling roots and hypocotyls were collected for RNA extraction, followed by gene expression analysis. The entire experiment was repeated three times, each involving a minimum of 20 seedlings from each line (WT and *Slalkbh10b* mutant).

### 4.7. Drought and Salt Tolerance Assays

Thirty seedlings—one-month-old, and exhibiting consistent growth from both WT and *Slalkbh10b* mutant lines, were selected and placed in trays of the same size. The trays were filled with water overnight to allow the plants to absorb sufficient moisture. Subsequently, the excess water was drained and no further irrigation was administered. Once the soil moisture content and humidity attained a moderate level, photographs of plants were taken, and this stage was recorded as Day 0 for both the WT and knockout lines. Before the drought treatment, the leaves relative water loss rate was assessed. Thirty leaves from each WT and *Slalkbh10b* mutant line were selected and placed upon dry filter paper. At regular intervals of 2 h, spanning a total duration of 36 h, the leaves were continuously weighed, thereby facilitating the calculation of the relative water loss rates. For the drought tolerance assay, the WT and *Slalkbh10b* mutant lines were cultivated in a greenhouse for 15 days without supplementary irrigation. Conversely, for the salt stress experiments, the plants were irrigated with water containing 300 mM NaCl (100 mL) every two days for a period of 20 days. Before and after treatment, leaf samples of WT and *Slalkbh10b* mutant lines were collected and used in subsequent analyses. Three biological replicates were conducted.

### 4.8. Determination of Physiological Indexs

Leaves at corresponding stages of development and positions were collected from WT and *Slalkbh10b* plants before and after exposure to drought and salt stress. Subsequently, we assessed their relative water content (RWC) [43], relative electric conductivity (REC) [44], hydrogen peroxide (H_2_O_2_) [45], malondialdehyde (MDA) [36], and proline [46] levels using established protocols. The determination of soluble sugar and starch contents was conducted in accordance with our previous study [47]. The chlorophyll content in mature leaves was determined using a previously reported methodology [48]. Additionally, in the trypan blue staining experiment, leaf samples were immersed in a trypan blue staining solution (50 mg trypan blue powder dissolved in 50 mL double-distilled water), gently agitated (50 rpm) for 4 h, and then heated in boiling water for 10-min. Subsequently, the samples were indirectly heated in 95% ethanol for 10-min before capturing images.

### 4.9. RNA Isolation and Quantitative Real-Time PCR

The frozen tomato tissues were ground into a fine powder using liquid nitrogen. Next, 0.2 g of the powdered tissue was placed in a 1.5 mL centrifuge tube with 1 mL of TRIzol. The subsequent steps were carried out according to the instructions provided in the RNA extraction kit (Invitrogen, Shanghai, China). The quality and quantity of the total RNA were assessed by measuring the absorbance ratios (OD 260/280 and OD 260/230) and performing 1.5% (*v*/*v*) agar gel electrophoresis. The first cDNA strand was synthesized using M-MLV reverse transcriptase (Promega, Beijing, China). The synthesized cDNA (20 μL) was diluted with two times the volume of RNase/DNase-free water for qRT-PCR reactions (5 μL 2 × GoTaq^®^ qPCR Master Mix enzyme, 3.5 μL nuclease-free water, 0.5 μL primers, 1 μL cDNA after dilution). The qRT-PCR analysis was conducted using the CFX96 Touch™ Real-Time PCR Detection System (Bio-Rad, Hercules, CA, USA). The PCR amplification parameters were set: an initial denaturation step at 95 °C for 2-min, followed by 40 amplification cycles (95 °C for 15 s and 60 °C for 40 s). After the qRT-PCR cycles, a melting curve analysis was performed to examine the specificity of the primers. The tomato *SlCAC* gene (Solyc08g006960) was utilized as an internal reference to normalize gene expression in tomato [49]. The relative gene expression levels were analyzed using the 2^−ΔΔCT^ method [50]. Each experiment was performed in triplicate, with three biological replicates and three technical replicates.

### 4.10. Primers and Accession Numbers

All gene-specific primers utilized in this study are shown in Appendix A, while those used in vector construction are shown in Appendix A.

### 4.11. Statistical Analysis

The data presented are three independent experiments’ mean ± standard error (SE). IBM SPSS Statistics 20 software (IBM Corp., Armonk, NY, USA) was used for significance analysis. Multiple comparisons were assessed via the Duncan test, and diverse lowercase indicated statistically significant differences (*p* < 0.05).

## 5. Conclusions

Among the ten ALKBH family proteins in tomato (*Solanum lycopersicum*), five of them can be classified into ALKBH9 and ALKBH10 subfamilies and share homology with HsALKBH5. Notably, the coding gene of SlALKBH10B exhibits a wide range of tissue expression patterns. Moreover, exogenous hormone ABA and drought and salt stress treatments induce its expression. SlALKBH10B is localized in the nucleus and cytoplasm, consistent with the subcellular localization of ALKBH10B in Arabidopsis and cotton. The knockout of *SlALKBH10B* enhances the sensitivity of mutants to ABA treatment, resulting in an upregulation of gene expression associated with ABA synthesis and response. The *Slalkbh10b* mutants show improved tolerance to drought and salt stress, with phenotypic differences in water retention, accumulation of photosynthetic products, accumulation of proline, accumulation of reactive oxygen species, and cellular damage. Collectively, our research outcomes shed light on the negative regulatory role of *SlALKBH10B* in drought and salt tolerance. These findings contribute to a comprehensive understanding of the biological functionality of the SlALKBH10B gene under abiotic stresses and provide novel genetic resources for the development of drought and salt-tolerant tomato cultivars.

## Figures and Tables

**Figure 1 ijms-25-00173-f001:**
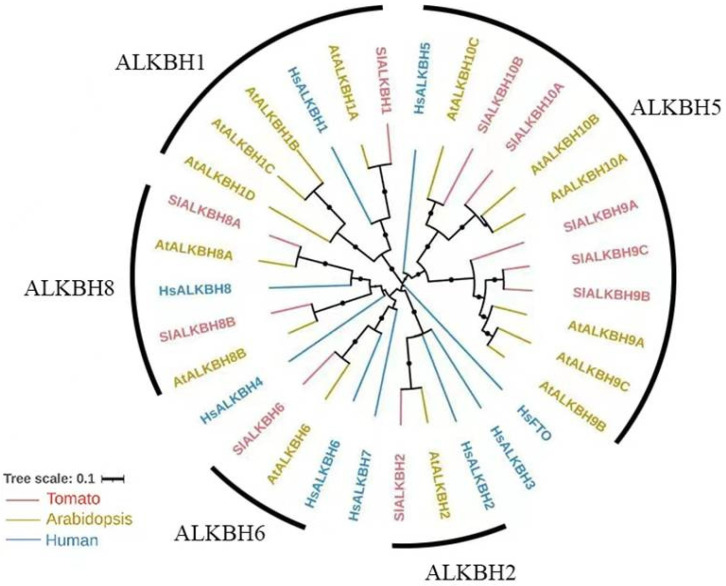
Phylogenetic analysis of the ALKBH family in tomato. Hs: *Homo sapiens*; At: *Arabidopsis thaliana*; Sl: *Solanum lycopersicum* (tomato).

**Figure 2 ijms-25-00173-f002:**
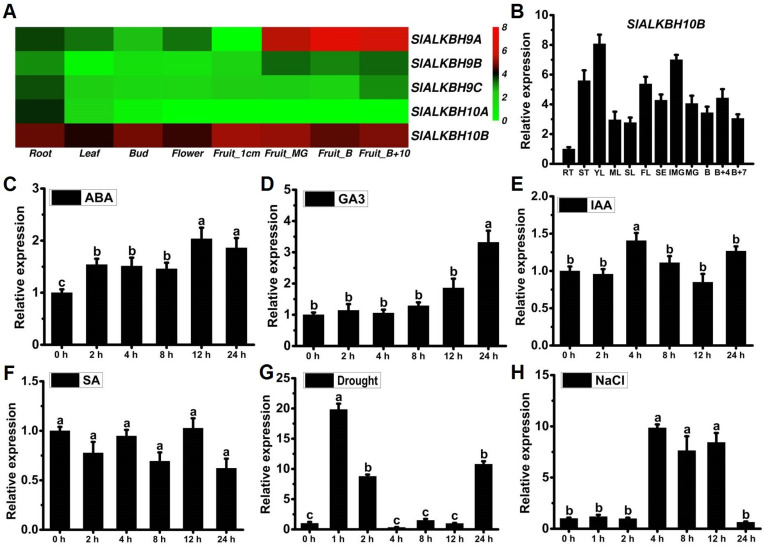
Expression analyses of *SlALKBH10B*. (**A**) Expression levels of tomato ALKBH9 and ALKBH10 genes based on transcriptome expression data (FPKM). Each column represents a different tissue at different developmental stages of tomato. The bar on the right indicates normalized expression data from high to low (red to green). (**B**) The relative expression of *SlALKBH10B* across different tissues detected by qRT-PCR. RT, root; ST, stem; YL, young leaves; ML, mature leaves; SL, senescent leaves; FL, flower; SE, sepe; IMG, immature green; MG, mature green; B, breaker stage; B + 4, four days after breaker stage; B + 7, seven days after breaker stage. (**C**–**F**) The relative expression of *SlALKBH10B* following exogenous treatment with abscisic acid (ABA) (**C**), gibberellin (GA_3_) (**D**), auxin (IAA) (**E**), and salicylic acid (SA) (**F**). (**G**,**H**) The relative expression of *SlALKBH10B* under drought (**G**) and salt (**H**) treatment. Each value represents the mean ± SE of three biological replicates. Multiple comparisons were assessed via the Duncan test, and diverse lowercase indicated statistically significant differences (*p* < 0.05).

**Figure 3 ijms-25-00173-f003:**
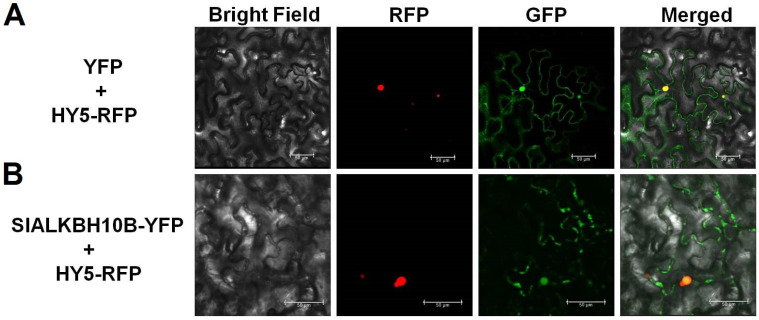
Subcellular localization of SlALKBH10B. (**A**) Subcellular localization of YFP proteins in *N*. *benthamiana* leaves. (**B**) Subcellular localization of SlALKBH10B-YFP fusion proteins in *N*. *benthamiana* leaves. HY5-RFP fusion proteins were used as the nuclear signal. The red fluorescence signal is elicited by RFP (red fluorescence protein) excitation, while the green fluorescence signal is induced by YFP (yellow fluorescence protein) excitation. The yellow signal resulted from the convergence of the red and green fluorescence signals. Scale bar = 50 μm.

**Figure 4 ijms-25-00173-f004:**
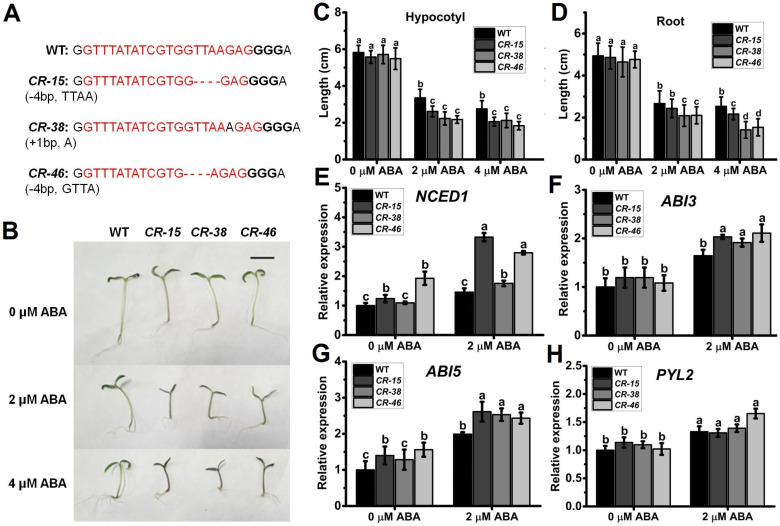
Analysis of ABA sensitivity. (**A**) The genotypes of *Slalkbh10b* mutant lines (*CR-15*, *CR-38* and *CR-46*). The target sequence on the genome is represented in red font, followed by the PAM sequence “GGG”. WT, wild-type. (**B**) Phenotypes of WT and *Slalkbh10b* mutants seedlings treatment with ABA. Scale bar = 2 cm. (**C**,**D**) Hypocotyl (**C**) and root (**D**) lengths of seedlings under control and ABA conditions, respectively. Each value represents the mean ± SE (*n* = 20) of three biological replicates. (**E**–**H**) The relative expression of *NCED1* (**E**), *ABI3* (**F**), *ABI5* (**G**), and *PYL2* (**H**) in WT and *Slalkbh10b* mutants seedlings treatment with or without 2 μM ABA. Each value represents the mean ± SE of three biological replicates. Multiple comparisons were assessed via the Duncan test, and diverse lowercase indicated statistically significant differences (*p* < 0.05).

**Figure 5 ijms-25-00173-f005:**
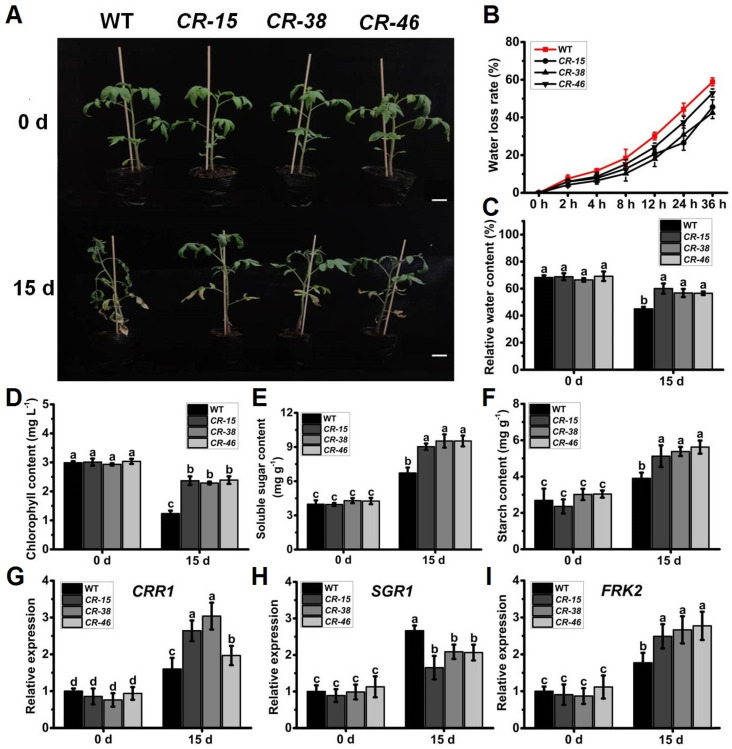
*Slalkbh10b* plants exhibited increased drought tolerance. (**A**) Phenotypes of WT and *Slalkbh10b* plants under drought treatment. Scale bar = 2 cm. (**B**) Water loss rate of detached leaves. (**C**) Relative water content. (**D**–**F**) The content of chlorophyll (**D**), soluble sugar (**E**), and starch (**F**). (**G**–**I**) The relative expression of *CRR1* (**G**), *SGR1* (**H**), and *FRK2* (**I**) in WT and *Slalkbh10b* leaves. Each value represents the mean ± SE of three biological replicates. Multiple comparisons were assessed via the Duncan test, and diverse lowercase indicated statistically significant differences (*p* < 0.05).

**Figure 6 ijms-25-00173-f006:**
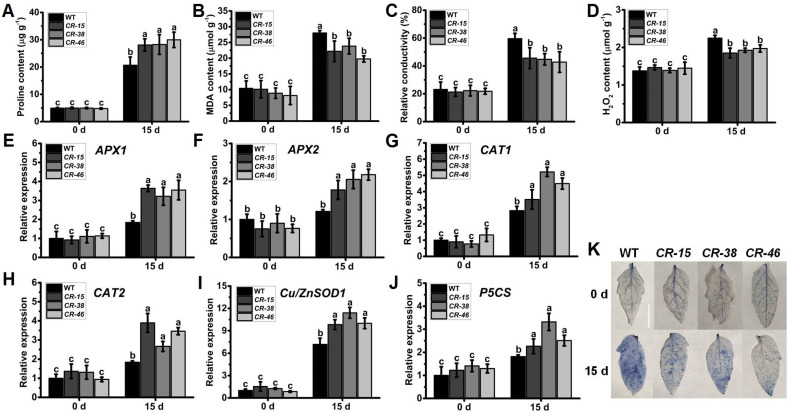
*Slalkbh10b* plants exhibited decreased cell damage after drought stress. (**A**–**C**) The content of free proline content (**A**), malondialdehyde (MDA) (**B**), and hydrogen peroxide (H_2_O_2_) (**C**). (**D**) Relative electric conductivity (REC). (**E**–**J**) The relative expression of cytosolic ascorbate peroxidase (APX) gene, *APX1* (**E**) and *APX2* (**F**), catalase gene, *CAT1* (**G**) and *CAT2* (**H**), superoxide dismutase [Cu-Zn] gene, *Cu/ZnSOD1* (**I**), and proline synthesis gene, *P5CS* (**J**) in WT and *Slalkbh10b* leaves. (**K**) Trypan blue staining. Scale bar = 2 cm. Each value represents the mean ± SE of three biological replicates. Multiple comparisons were assessed via the Duncan test, and diverse lowercase indicated statistically significant differences (*p* < 0.05).

**Figure 7 ijms-25-00173-f007:**
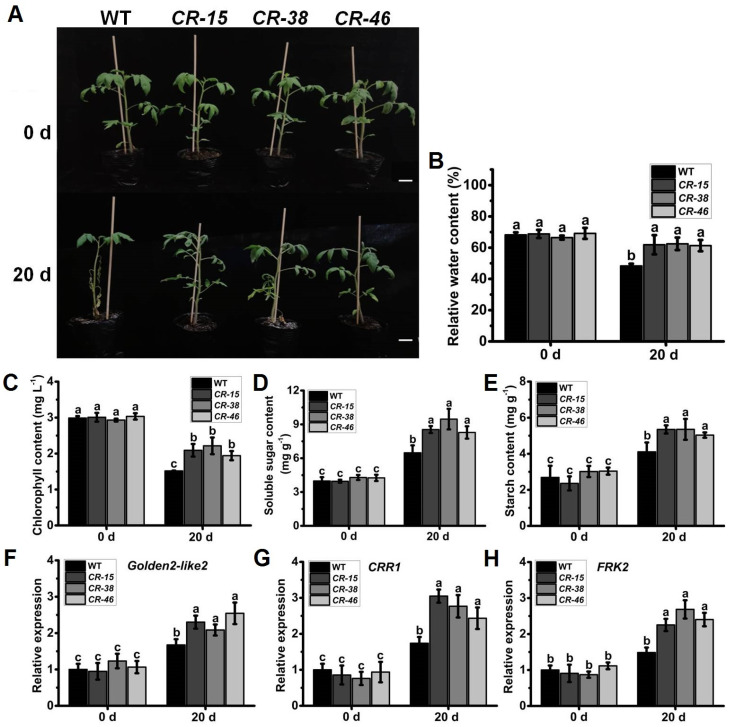
*Slalkbh10b* plants exhibited increased salt tolerance. (**A**) Phenotypes of WT and Slalkbh10b plants under drought treatment. Scale bar = 2 cm. (**B**) Relative water content. (**C**–**E**) The content of chlorophyll (**C**), soluble sugar (**D**), and starch (**E**). (**F**–**H**) The relative expression of *Golden2-like2* (**F**), *CRR1* (**G**), and *FRK2* (**H**) in WT and *Slalkbh10b* leaves. Each value represents the mean ± SE of three biological replicates. Multiple comparisons were assessed via the Duncan test, and diverse lowercase indicated statistically significant differences (*p* < 0.05).

**Figure 8 ijms-25-00173-f008:**
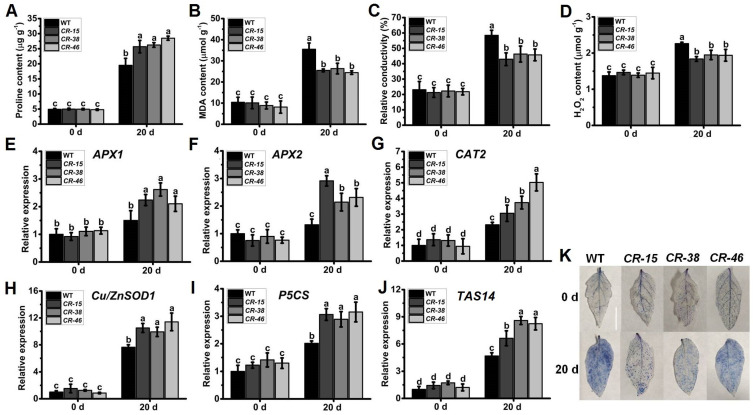
*Slalkbh10b* plants exhibited decreased cell damage after salt stress. (**A**–**C**) The content of free proline content (**A**), malondialdehyde (MDA) (**B**), and hydrogen peroxide (H_2_O_2_) (**C**). (**D**) Relative electric conductivity (REC). (**E**–**J**) The relative expression of cytosolic ascorbate peroxidase (APX) gene, *APX1* (**E**) and *APX2* (**F**), catalase gene, *CAT2* (**G**), superoxide dismutase [Cu-Zn] gene, *Cu/ZnSOD1* (**H**), proline synthesis gene, *P5CS* (**I**), and dehydrin gene, *TAS14* (**J**) in WT and *Slalkbh10b* leaves. (**K**) Trypan blue staining. Scale bar = 2 cm. Each value represents the mean ± SE of three biological replicates. Multiple comparisons were assessed via the Duncan test, and diverse lowercase indicated statistically significant differences (*p* < 0.05).

## Data Availability

The data that support the findings of this study are available from the corresponding author upon reasonable request.

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
