# Peer review of "The AlkB Homolog SlALKBH10B Negatively Affects Drought and Salt Tolerance in Solanum lycopersicum"

_ijms, 2023, doi:10.3390/ijms25010173_

Round 1
Reviewer 1 Report
Comments and Suggestions for Authors
Line 13: which plants? Please, specify.
Line 15: in many plants? Which?
Lines 63- 63: please, edit grammar to avoid repetition.
Line 90: “during the tomato seedling stage“ ?? Do you mean at seedling stage, not during?
Line 91: „morphological, statistical, biochemical, and molecular analyses in Slalkbh10b mutant lines“: not „in“ but „of“; Statistical analysis itself is not exist. Statistacal can be apply to morpholocial analysis to confirm significance etc. How can you separate biochemical and molecular analysis? It is largely the same.
Lines 139 – 149: please, provide information what do you mean as basic level (expression 1). I expect that it is level in the control (0h), but it is not look like this..
Line 153: citation is missing.
Line 158: „expressed fluorescently in the entirety of the cell“ ?? localised or fluoresence has been detected. Moreover, it is well-known that localization is dependent form cell type. Current results only confirmed protein localization in Nicotiana leaf cell, but not general one. In the other cell type protein can localise in different places. Please, also provide details of microscope, wavelength etc in M&M.
Fig 5. Bright field is too dark and did not provide usefull information. Please, either provide SR2200 label or change brigthness of first column.
Line 190: „(CR-15, CR-38 and CR-46) after gene editing“ : CR lines already means gene editing, „after gene editing“ is redundant.
Lines 340 and 357 are similar. Please, edit.
Figure 8: what is APX1 and APX2? Cat1 and Cat2? What is SOD? Cytoplamsic Cu/Zn? Plastid Cu/Zn, Fe- SOD? Etc? Please, provide information which level you have used as basic. What was the house-keeping gene in qPCR?
Line 457: I am not sure spraying is an optimal method: how do you know what was the real concentration of the hormones in the tissue? Did you study which concentration you used is specific? It may happens that 100 µM ABA is not a specific, because in seedlings you have used apr. 10 fold less ABA… Similar question is for IAA, GA , SA.
Line 499: what is the reason to perform treatment with 3 mM Chlorine (Cl -)? Did you add sucrose to the medium? How many? What is light intensity?
Line 500: not mM, but µM (mkM). Probably, technical mistake during pdf conversion.
7 days treatment provide you adaptation mechanism, not response one. For response you need to study only first 3-12 hours.
Line 546: after dilution.
Comments on the Quality of English Languagesome sentences required corrections.
Reviewer 2 Report
Comments and Suggestions for Authors
The research results are interesting and make a new contribution to the development of science. Nevertheless, I have a few comments:
1) The research hypothesis and purpose of the study are missing - please complete.
2) L 153: No literature citation in brackets.
3) I suggest enriching all graphs with homogeneous groups defined by Duncan's test, within which the compared averages are not significantly different (as was done in Figrure 2).
4) Since the research on plant response to drought and salinity was carried out on two dates (0d and 15d) (Figure 5-8), it is worth comparing the results of the research not only within each measurement date, but also between dates, based on a two-way ANOVA with repeated measurements (evaluation of time as a factor).
5) The readability of the figures can be improved.
Round 2
Reviewer 1 Report
Comments and Suggestions for Authors
Thank you for clear explanations.
There are few points need to be pointed out in the final version.
Figure 8, panels E - H: there is not APX1, APX2 or SOD, ROS act (produced, induved signaling, damage mebrane) only in local places (plasma membrane, chloroplasts, mitochondria). That's why plants have cytolamic APX and stroma APX whuch have a different function, cyt and plastid SOD etc. Please, provide link to localization/function.
Comments 16: 7 days treatment provide you adaptation mechanism, not response one. For response
you need to study only first 3-12 hours.
Response 16: Indeed, I concur with your perspective regarding adaptation and responsiveness. In our experiment, we encountered challenges in sampling sterilized seeds within the initial 3-12 hour period. As a result, we opted to subject the seeds to abiotic stress treatment for a duration of 7 days. Based on our experience, we have found that after 7 days of cultivation, the seeds typically reach a stage where they have just emerged from the
cotyledon. This growth stage is particularly convenient for us to observe the overall growth state of the seedlings and facilitates the sampling process for detecting internal changes.
You are right, it isnit so easy on seeds stage (as tomato) to do synhronization. But anyway, after 7 days plants shown "adaptive" response with many pathways are changed. And you can not know what is the primary signal, what is only adaptation. Please, only mention this in the text.
Line 541: there is 3 mM halogen (as Cl ion) in your substrate what is consider as treatments with chemical regulator.
Comments on the Quality of English Languagesome small polishing are require during proof reading.
Reviewer 2 Report
Comments and Suggestions for Authors
The Authors responded to most of my comments and made appropriate corrections to the manuscript. In the review, I suggested additional statistical analysis of the experimental results, which would broaden the possibilities of inference. Since the study of plant response to drought and salinity was conducted on two dates (0d and 15d) (Figure 5-8), I suggested comparing the results of the study not only within each measurement date, but also between dates, based on a two-way ANOVA with repeated measurements (evaluating time as a factor). In response to comments, the authors confirm performing such a statistical analysis, but did not actually do it. Nevertheless, I believe that the scientific value of the article has been raised ompared to the original version, and in this form it can be published in the International Journal of Molecular Sciences.
Author Response
Thank you so much for taking the time out of your busy schedule to review this manuscript. Your valuable comments are crucial in improving its quality. Thanks again!